# Enhancement in Sensitivity and Selectivity of Electrochemical Technique with CuO/g-C_3_N_4_ Nanocomposite Combined with Molecularly Imprinted Polymer for Melamine Detection

**DOI:** 10.3390/polym16131800

**Published:** 2024-06-25

**Authors:** Dalawan Limthin, Piyawan Leepheng, Benchapol Tunhoo, Annop Klamchuen, Songwut Suramitr, Thutiyaporn Thiwawong, Darinee Phromyothin

**Affiliations:** 1College of Materials Innovation and Technology, King Mongkut’s Institute of Technology Ladkrabang, Bangkok 10520, Thailandmildpiyawan55@gmail.com (P.L.); benchapol.tu@kmitl.ac.th (B.T.); thutiyaporn.th@kmitl.ac.th (T.T.); 2Electronic and Control System for Nanodevice Research Laboratory (ECSN), College of Materials Innovation and Technology, King Mongkut’s Institute of Technology Ladkrabang, Bangkok 10520, Thailand; 3National Nanotechnology Center, National Science and Technology Development Agency, Patumthani 12120, Thailand; annop@nanotec.or.th; 4Department of Chemical, Faculty of Science, Kasetsart University, Ladyao, Chatuchak, Bangkok 10900, Thailand; fsciswsm@ku.ac.th

**Keywords:** CuO/g-C_3_N_4_ nanocomposite, molecularly imprinted polymer, melamine, photoelectrochemical

## Abstract

This study focused on enhancing the sensitivity and selectivity to detect melamine by utilizing a photoelectrochemical method. This was achieved by combining a melamine-imprinted polymer with a CuO/g-C_3_N_4_ nanocomposite, which was synthesized through chemical precipitation and calcination. The resulting nanocomposite exhibits improved carrier mobility and photoelectrochemical properties. A molecularly imprinted receptor for selective detection was created through bulk polymerization with methacrylic acid and a melamine template. The characterization of the nanocomposite was performed using X-ray photoelectron spectroscopy for the chemical oxidation state, X-ray diffraction patterns for the crystalline structure, and ultraviolet/visible/near-infrared spectroscopy for optical properties. The CuO/g-C_3_N_4_ nanocomposite exhibits photoactivity under visible light. The modified electrode, incorporating the CuO/g-C_3_N_4_ nanocomposite and melamine-imprinted polymer, demonstrates a linear detection range of 2.5 to 50 nM, a sensitivity of 4.172 nA/nM for melamine, and a low detection limit of 0.42 nM. It shows good reproducibility and high selectivity to melamine, proving effective against interferences and real samples, showcasing the benefits of the molecularly imprinted polymer.

## 1. Introduction

Melamine is an organic compound identified by its heterocyclic triazine structure, which is abundant in nitrogen [1]. It is extensively utilized in manufacturing plastic goods, adhesives, fertilizers, food packaging, and laminated products, including fire-retardant coatings, cleaning supplies, and cement. The appeal of melamine lies in its cost-effectiveness and superior performance characteristics. However, the widespread application of melamine raises concerns about potential contamination in food items. This contamination poses health risks, including symptoms such as hematuria, kidney stones, and, in severe cases, kidney failure, particularly in cases of overdose, where ingestion can be fatal [2]. Several European member states have banned melamine products from being used in food-related items. The health consequences and toxicological properties of cyanuric acid and melamine were analyzed in a report by an expert meeting organized by the World Health Organization (WHO) [3]. The report recommended tolerable daily intake (TDI) levels of 0.2 mg/kg of body weight for melamine and 1.5 mg/kg for cyanuric acid, respectively [1,2,4]. Consequently, the imperative for the swift and precise detection of melamine has become paramount in ensuring safety and health standards.

Melamine detection is commonly conducted through various techniques, including high-performance liquid chromatography (HPLC) [5,6], gas chromatography/mass spectrometry (GC/MS) [7,8], surface-enhanced Raman spectroscopy [9], and fluorescence spectroscopy [10]. Despite offering excellent selectivity, sensitivity, and accuracy, these methods are characterized by their complexity, high cost, time-intensive procedures, and the need for specialized operations. In recent times, several research studies have explored photoelectrochemical techniques, integrating photoactive materials and electrochemistry for sensor application [11]. This technique has several advantages: simplicity, high sensitivity, rapid response times, low detection limit, cost-effectiveness, and disposability [11,12,13]. Moreover, it has the potential to be transformed into a portable device, enhancing its versatility and convenience [14,15].

Graphitic carbon nitride (g-C_3_N_4_) is categorized as an n-type organic semiconductor characterized by a conjugated polymer with a triazine ring structure [16]. One of its notable characteristics is its ability to function as a photocatalyst, with a significant band gap of 2 to 3 eV, allowing it to be active in visible light [17]. Additionally, it is characterized by being a metal-free material, exhibiting good chemical and electronic structure stability, possessing a large specific surface area, and being cost-effective and environmentally friendly [16,18]. Graphitic carbon nitride has been extensively studied in various research fields, such as solar cells, energy conversion, light-emitting diode (LED) fabrication, sensing technology, pollutant degradation, imaging applications, and photocatalytic water splitting [19,20]. Nevertheless, graphitic carbon nitride has the drawback of displaying limited carrier mobility, inefficiencies in charge transfer, and rapid charge recombination [21,22]. These limitations can be effectively addressed through the coupling of semiconductor materials, offering a promising solution to enhance the performance of graphitic carbon nitride and reduce recombination rates [22].

Recent studies have shown that g-C_3_N_4_ can be combined with other semiconductors, such as Fe_3_O_4_ [23,24], CdS [25,26], ZnO [27,28], TiO_2_ [29], and CuO [30,31,32,33]. This coupling technique aims to enhance the separation efficiency of photogenerated electron/hole pairs, hence improving photocatalytic activity [19,21,22]. ZnO and TiO_2_ are characterized by a relatively wide band gap, ranging from 3 to 3.2 eV [34]. A higher energy input must be required, specifically in photoenergy, to produce electron/hole pairs. The wide band gap influences their absorption characteristics, making them more responsive to higher-energy photons, usually in the ultraviolet range, and less sensitive to lower-energy photons, such as those in the visible light spectrum [22,27,35]. Copper oxide has been widely used in numerous research studies, including photocatalysts, solar cells, supercapacitors, sensing technologies, gas sensors, and biomedical applications. Its monoclinic crystal structure makes it a p-type semiconductor, and it is well known for its photoactive properties. Owing to its narrow band gap, which ranges from 1.2 to 1.7 eV, this material is sensitive to visible and infrared radiation [36,37,38]. Additionally, it is known for being low-cost, abundant, easily preparable, and environmentally friendly. The combination of CuO and g-C_3_N_4_, known as CuO/g-C_3_N_4_, forming a p–n heterojunction, enhances charge carriers through photoactive excitation under visible light. This synergistic combination holds promise for sensitivity enhancement in photoelectrochemical detection [22,30,33].

The efficiency of biosensors greatly depends on their selectivity performance [39]. Recent studies have emphasized the importance of biosensor receptors, including enzymes, proteins, antigens, antibodies, and aptamers. Although natural receptors have a high affinity for targets and fast response times, they frequently experience instability in chemical and physical conditions, limited heat and pH resistance, challenges in preparation, and high expenses [39,40,41].

On the other hand, molecularly imprinted polymers (MIPs) represent a type of polymer matrix receptor designed with a specific cavity imprinted for a target molecule [42]. MIPs provide a strong binding capacity because of their precisely tailored cavities that perfectly match the desired molecule’s size, shape, and functional groups. This cavity facilitates interactions through hydrogen bonding, as well as covalent and non-covalent bonding. MIPs offer high affinity and demonstrate robustness against physical and chemical conditions. They are easily synthesized, cost-effective, and resistant to fluctuations in temperature and pH [43,44,45]. The molecular imprinting technique has recently gained considerable attention due to its wide range of applications, such as food allergy detection [46,47,48], drug delivery [49], medicinal applications [50], biosensors [51], pesticides [52,53], and others.

In this work, we developed a melamine detection system utilizing a CuO/g-C_3_N_4_ nanocomposite combined with a molecularly imprinted polymer, and detection was carried out using the photoelectrochemical technique. The CuO/g-C_3_N_4_ composite was fabricated using the chemical precipitation and calcination synthesis techniques. The analysis of the CuO/g-C_3_N_4_ composite involved X-ray photoelectron spectroscopy, X-ray diffraction patterns, and ultraviolet/visible/near-infrared spectroscopy. These techniques provided information about the composite’s crystalline structure, chemical oxidation state, and optical characteristics. Subsequently, the CuO/g-C_3_N_4_ nanocomposite, in combination with the melamine-imprinted polymer, was applied as a modification on the carbon plate electrode. Melamine was utilized as the template, and methacrylic acid was employed as the monomer to polymerize, forming the melamine-imprinted polymer. The modified electrode was performed using photoelectrochemical techniques under visible light, and measurements were conducted using the cyclic voltammetry (CV), differential pulse voltammetry (DPV), and amperometry modes (AMP) of detection. The melamine detection performance was effectively demonstrated through electrode surface modification using a molecularly imprinted polymer in conjunction with the CuO/g-C_3_N_4_ nanocomposite. The achieved results included sensitivity, a low detection limit, a dynamic range, reproducibility, and specificity towards melamine.

## 2. Materials and Methods

### 2.1. Materials

The following chemicals were procured from the Sigma-Aldrich company (St. Louis, MA, USA): copper (II) nitrate trihydrate, ethylene glycol dimethacrylate (EGDMA, 98%), phosphate buffer saline (PBS, pH 7.4), methacrylic acid (MAA, 99%), 0.2 M 2,2-azobisisobutyronitrile in toluene, thiourea (AIBN, 99%), urea (99%), uric acid (99%), and ammonia (25%). Carlos Company (Cornaredo, Italy) supplied melamine powder 99%, ethanol 99.8%, chloroform 99.5%, glycine 99%, and sodium hydroxide pellets for analysis. All of the chemicals used were of analytical reagent grade. In the actual sample analysis, we obtained three types of milk—cow milk, strawberry milk, and almond milk—from a local market in Bangkok, Thailand.

### 2.2. Preparation of CuO/g-C_3_N_4_ Nanocomposite

The production of graphitic carbon nitride (g-C_3_N_4_) was achieved by subjecting melamine to a one-step calcination process. Initially, 100 g of melamine was subjected to calcination in a partially enclosed crucible at a temperature of 550 °C for a duration of 4 h. This process was carried out in the presence of air, and the rate of heating was set at 5 °C/min. During this process, the powder transformed from white to pale-yellow powder. Subsequently, the synthesized sample was placed in a desiccator for storage. Subsequently, CuO/g-C_3_N_4_ was synthesized using the chemical technique, as seen in Figure 1a. Initially, a solution was prepared by mixing 0.1 M copper (II) nitrate trihydrate, serving as the precursor, with 25 g of graphitic carbon nitride in 100 mL of deionized water. Subsequently, 1 M NaOH solution was gradually added to the mixture while continuously stirring for 2 h until the pH level reached 14. Remarkably, the original blue color of the solution transitioned to black during this stage. Next, the sample solution was collected and underwent multiple treatments using deionized water until the pH reached a stable value of 7. The precipitates were dried at 100 °C after being thoroughly washed. The nanoparticle product obtained was subjected to calcination at a temperature of 350 °C for a duration of 4 h. Finally, the synthesized CuO/g-C_3_N_4_ nanoparticle was stored in a dry desiccator, completing the fabrication process. The CuO/g-C_3_N_4_ electrode was applied to the carbon plate electrode (CPE). A solution containing 5 mg/mL CuO/g-C_3_N_4_ in a 1% MAA solution was dropped onto the CPE. Subsequently, the modified electrode was subjected to a 30 min curing process under UV light.

### 2.3. Fabrication of CuO/g-C_3_N_4_ Nanocomposite Combined with Molecularly Imprinted Polymer

The CuO/g-C_3_N_4_ electrode was modified on the carbon plate electrode, combining a nanocomposite of CuO/g-C_3_N_4_ with a melamine-imprinted polymer. Bulk polymerization was utilized in the synthesis, incorporating CuO/g-C_3_N_4_, melamine, MAA, AIBN, and EGDMA as a nanomaterial, template, monomer, initiator, and crosslinker, respectively. Figure 1b illustrates the procedure for modifying the electrode surface. A solution was prepared by combining 1 mL of MAA dissolved in 2.5 mL chloroform with 2 mL of 1 mM melamine. Next, AIBN and EGDMA were added to the mixture in a molar ratio of 1.0:0.5:5.0. Subsequently, a solution containing 5 mg/mL CuO/g-C_3_N_4_ was incorporated into the mixture, followed by 30 min of stirring at 60 °C. The sensing region was determined using laminated Kapton tape with a diameter of 3 mm. Subsequently, it was subjected to 30 min of UV irradiation, resulting in the formation of a polymer matrix on the electrode. The CuO/g-C_3_N_4_/MIP electrode template was then eluted multiple times with deionized water and ethanol. Finally, the CuO/g-C_3_N_4_/MIP electrode was dried and kept in a desiccator at room temperature.

This study investigated four different surface-modified electrodes, as shown in Figure 1b: the bare carbon plate electrode (CPE), CuO/g-C_3_N_4_ modified on the CPE (CuO/g-C_3_N_4_/CPE), CuO/g-C_3_N_4_ mixed with melamine-imprinted polymers (CuO/g-C_3_N_4_/MIP), and CuO/g-C_3_N_4_ combined with non-imprinted polymers (CuO/g-C_3_N_4_/NIP). Non-imprinted polymers (NIPs) were synthesized using a method similar to molecularly imprinted polymers (MIPs) but without the presence of a template molecule.

### 2.4. Material Characterizations

The CuO/g-C_3_N_4_ nanocomposite’s crystalline structure was examined using X-ray diffraction (XRD) with a Rigaku SmartLab apparatus equipped with Cu Kα radiation (wavelength of 1.5406 Å). X-ray photoelectron spectroscopy (XPS) was employed to ascertain the elemental composition of the surface of a material. Furthermore, it offers information regarding the binding state of the existing elements. X-ray photoelectron spectroscopy (XPS) is a great technique for comprehending the chemical composition, oxidation states, and electronic surroundings of the elements present on the surface of a CuO/g-C_3_N_4_ nanocomposite. The optical characteristics were determined using a UV-VIS-NIR spectrophotometer. An atomic force microscope (AFM) was used to evaluate the surface modification of electrodes made of a CuO/g-C_3_N_4_ nanocomposite combined with molecularly imprinted polymer electrodes (CuO/g-C_3_N_4_/MIP). In addition, a field emission scanning electron microscope with an Oxford energy-dispersive X-ray spectrometer (FESEM/EDS) was used to verify morphology and elemental composition on the surface modification of the modified electrode (CuO/g-C_3_N_4_/MIP electrode). Electrochemical measurements were detected using a potentiostat (μSTAT 400, Dropsens, Oviedo, Spain) that was controlled by Dropview 8400 software, version 2.2 15B1204.

### 2.5. Electrochemical Measurements

The performance of melamine detection was evaluated using electrochemical testing, specifically measured using cyclic voltammetry, differential pulse voltammetry, and amperometry mode. The carbon paste electrode with a diameter of 3 mm was employed in a tri-electrode configuration. The counter and reference electrodes were constructed using carbon paste and silver/silver chloride, respectively. To study the photoelectrochemical behavior of the sensing layer, the device was illuminated with a Nikon Corporation (model C-FI 115, Nissei Electroic Co., Ltd., Osaka, Japan) quartz tungsten-halogen lamp with a set light intensity of approximately 3 mW/cm^2^. Cyclic voltammetry (CV) was used to measure the electrode potentials of different electrodes in a PBS buffer. The buffer contained a potassium ferrocyanide solution with a concentration of 50.0 mM and a pH of 7.4 at 25 °C. The applied voltage ranged from −1.0 V to +1.0 V. Furthermore, CuO/g-C_3_N_4_/MIP was examined using differential pulse voltammetry (DPV) in a PBS buffer solution with varied concentrations of melamine. The observation was conducted at a scan rate of 20.0 mV/s, a pulse width of 50.0 ms, and a pulse amplitude of 50.0 mV. The potential range for this observation was between −0.1 V and −0.3 V. Electrodes modified with CuO/g-C_3_N_4_/MIP were assessed for their efficacy utilizing amperometry techniques (AMP). This involved maintaining a steady current at −0.2 voltage and defining the linear calibration curve, sensitivity, and detection limit. For the selectivity study, interference comparisons were conducted for each of thiourea, ammonia, glycine, urea, uric acid, and melamine solutions with consistent concentrations of 100.0 nM.

## 3. Results and Discussions

### 3.1. Characterization of CuO/g-C_3_N_4_ Nanocomposite Synthesis

The nanocomposite was synthesized using the co-precipitation and calcination techniques. Analytical techniques, including X-ray diffraction patterns, field emission scanning electron microscopy, X-ray photoelectron spectroscopy, and ultraviolet/visible/near-infrared spectroscopy, were used to gain knowledge about the material’s crystalline structure, morphology, chemical oxidation state, and optical properties.

As illustrated in Figure 2a–c, the surface morphology of the CuO/g-C_3_N_4_ nanocomposite was analyzed utilizing high-magnification field emission scanning electron microscopy (FESEM). The FESEM analysis revealed that pure g-C_3_N_4_ exhibited a lamellar structure, implying a layered or sheet-like structure due to its characteristics of a 2-dimensional nature. The structural configuration is a result of a conjugated polymer that contains a triazine ring structure within the g-C_3_N_4_ material [16]. Figure 2b, depicting the morphology of CuO, illustrates a sheet-like structure. Furthermore, it indicates that the thickness of the particles is less than the width dimension of sheet-like structures [54]. The CuO/g-C_3_N_4_ nanocomposites, as-prepared, exhibit a sheet-like structure. EDS mapping analysis revealed a dispersed distribution of C, N, Cu, and O elements on the surface of the CuO/g-C_3_N_4_ nanocomposites, confirming the successful combination of CuO and g-C_3_N_4_ in the material, as depicted in Figure 2d.

The XRD patterns were analyzed to determine the crystalline structures formed by the combination of g-C_3_N_4_ and CuO, as shown in Figure 3. The sample of graphitic carbon nitride displays distinctive diffraction peaks at 2θ = 13.06° and 27.5°. These peaks closely correspond to the crystallographic planes of g-C_3_N_4_, specifically (100) and (002). The observed peaks suggest the presence of interplanar stacking within the g-C_3_N_4_ structure, resulting from the presence of conjugated aromatic systems. These specific peaks observed in the data indicate the particular arrangement and stacking of heptazine units, which play a role in determining the crystallographic properties of the graphitic carbon nitride material [55].

The XRD pattern of CuO exhibited diffraction peaks at 2θ angles of 32.59°, 35.61°, 38.78°, 48.82°, 53.54°, 58.37°, 61.60°, 66.31°, and 68.15°. These peaks were identified as corresponding to specific crystallographic planes (hkl) that are characteristic of a monoclinic phase, according to the JCPDS standard no. 45-0937 for CuO nanomaterial. These planes present in this set are (110), (002), (111), (202), (020), (202), (113), (311), and (220) [56]. In order to confirm the synthesis of CuO/g-C_3_N_4_, the XRD pattern of the combined material should ideally exhibit diffraction peaks that correspond to both CuO and g-C_3_N_4_ phases. If the detected peaks of CuO in the XRD pattern coincide with the typical peaks of g-C_3_N_4_ (such as 27.5° for g-C_3_N_4_ and (002) and (111) for CuO), it would demonstrate the successful synthesis of CuO/g-C_3_N_4_. The confirmation of the composite structure can be achieved by observing the presence and overlap of distinctive peaks from both materials in the XRD pattern.

X-ray photoelectron spectroscopy (XPS) proves the oxidation states, chemical composition, and electronic environments of the elements present on the surface of a CuO/g-C_3_N_4_ nanocomposite. The XPS survey spectrum of CuO/g-C_3_N_4_ verified that the peaks correspond to the elements copper (Cu), carbon (C), oxygen (O), and nitrogen (N), as present in Figure 4a. Figure 4b–e illustrate the high-resolution spectra, also known as the core XPS spectra, which revealed the following elements: Cu 2p, O 1s, C 1s, and N 1s. Additionally, Cu 2p was identified in the high-resolution spectrum by two distinct peaks at 933.8 eV and 953.8 eV, corresponding to Cu 2p_3/2_ and Cu 2p_1/2_, respectively. The separation of the main peak positions of Cu 2p was calculated to be 20.0 eV, consistent with the findings documented in CuO spectra [57].

Furthermore, the confirmation of the CuO state was reinforced by observing broad satellite peaks at binding energies higher than those of the main peaks. Specifically, the presence of CuO is indicated by the presence of two satellite peaks on the higher binding energy side, approximately at 943.8 eV and 941.5 eV, which was close to the main peak of Cu 2p_3/2_ at 933.8 eV [58]. Notably, the separation of about 9.0 eV between the main peak of Cu 2p_1/2_ at 953.8 eV and its satellite peak at 962.5 eV further confirms the existence of CuO, as present in Figure 4b. The peaks at s529.7 eV in the O 1s spectrum indicate O atoms within Cu–O bonds. Conversely, the 531.7 and 532.7 eV peaks are ascribed to oxygen atoms originating from the -OH- groups on the CuO surface. The presence of an abundance of surface -OH- groups is a consequence of the existence of defect sites and oxygen vacancies on the surface of the catalyst [59], as illustrated in Figure 4c. The peak at 284.7 eV in the C 1s spectrum is definitively characterized as the C reference peak. The peak 286.1 eV corresponds to C-O, while the peak 288.5 eV results from sp2-bonded carbon (N-C=N) [60]. The C 1s binding energies at 288.5 eV are the interaction of g-C_3_N_4_ with CuO [61], as present in Figure 4d. Deconvoluting the N 1s spectrum of g-C_3_N_4_ revealed peaks at 398.4 eV, 400.1 eV, and 401 eV. These peaks correspond to the pyridine N, tertiary N, and graphitic N, respectively [60,61], as illustrated in Figure 4e.

Determining the optical band gap energy is a crucial aspect of optimizing semiconductor materials for applications involving light illumination. The band gap energy signifies the lowest amount needed to promote an electron’s transition from the valence band to the conduction band, generating electron/hole pairs. This process is fundamental in various semiconductor devices, such as solar cells and photodetectors, where efficient light absorption and charge carrier generation are essential for optimal performance. Comprehending and managing the optical band gap empowers engineers and researchers to customize materials for particular uses and improve the effectiveness of devices that depend on photo-induced processes [11,14]. The optical properties of the sample were evaluated by analyzing diffuse reflection. The energy band gap was calculated using the Kubelka–Munk relation, based on the theory proposed by P. Kubelka and F. Munk in 1931. This involved determining the x-axis intercept of the straight-line equation in the reflection spectra. This allowed for the calculation of the optical band gap through the Kubelka–Munk equation [62,63]
(1)$$F(R)=\frac{{\left(1-R\right)}^{2}}{2R}$$
where *R* represents the diffuse reflectance, illustrated in Figure 5a. The energy band gap values of CuO/g-C_3_N_4_ were determined using the application of the Kubelka–Munk relation and reported as 1.40 eV, 2.77 eV, and 1.44 eV, respectively. This indicates that a light source within the visible light can effectively power the prepared device. A band gap value of 1.44 eV suggests that the CuO/g-C_3_N_4_ nanocomposite was suitable for utilization with a light source in the visible range. This band gap value indicates that the material can effectively absorb photons in the visible light range, generating electron/hole pairs and facilitating various light-induced processes in potential devices.

A comparative analysis was conducted on the electrochemical characteristics of various materials, including CPE, CuO, g-C_3_N_4_, and CuO/g-C_3_N_4_. As illustrated in Figure 5b, these substances were fabricated on the carbon plate electrode and evaluated utilizing amperometry in a PBS solution at −0.2 voltage, both with and without light irradiation. Following light irradiation, the electrode composed of CuO/g-C_3_N_4_ demonstrated the highest negative current signal, followed by CuO, g-C_3_N_4_, and the bare electrode, respectively. Conversely, the cathodic current indication decreased when the light was turned off. This suggests that the presence or absence of light irradiation influenced the photoelectrochemical response observed in the electrodes. Therefore, the increase in the photoelectrochemical response during light irradiation can be attributed to the generation of electron/hole pairs with the CuO/g-C_3_N_4_ nanocomposite. The coupling of g-C_3_N_4_ with CuO reduced the recombination rate, ultimately enhancing carrier mobility and charge transfer efficiency in the electrode. The interaction between CuO and g-C_3_N_4_ is responsible for the improved photoelectrochemical characteristics of the nanocomposite.

### 3.2. A Surface-Modified Electrode with a CuO/g-C_3_N_4_ Nanocomposite Combined with a Molecularly Imprinted Polymer

CuO/g-C_3_N_4_/MIP was applied as a modification on a carbon plate electrode, which involved the combination of a CuO/g-C_3_N_4_ nanocomposite with a molecularly imprinted polymer. In this process, the molecularly imprinted polymer serves as a polymer matrix specifically designed with a cavity that matches the shape and properties of a particular molecule, referred to as the template. Subsequently, the template is later eluted, resulting in the formation of a specific cavity for the template molecule. This composite structure, consisting of CuO/g-C_3_N_4_ and a molecularly imprinted polymer, was designed for specific recognition and detection purposes, leveraging the selectivity provided by the imprinted polymer.

An atomic force microscope (AFM) was used to investigate the surface topography of the imprinted layer. In Figure 6, the atomic force microscopy (AFM) image displayed the morphology in 2D and 3D of the bare electrode, CuO/g-C_3_N_4_/NIP, and CuO/g-C_3_N_4_/MIP after the template removal. The AFM images clearly demonstrate a significant contrast in the surface roughness, which can be quantified using the root-mean-square (RMS) value. The RMS value is directly related to the roughness. The CuO/g-C_3_N_4_/NIP electrode had a surface roughness with a calculated RMS value of 31.59 nm, which was almost flat. On the other hand, the CuO/g-C_3_N_4_/MIP electrode had a more uneven surface with an RMS value of 149.64 nm. Compared to the CuO/g-C_3_N_4_/NIP electrode, the CuO/g-C_3_N_4_/MIP electrode’s AFM figure revealed more surface roughness, which was surface roughness due to the templates that were rinsed during the imprinting process, leaving behind certain recognition sites within the polymer matrix, cavities formed in the CuO/g-C_3_N_4_/MIP electrode, responsible for the increased surface roughness. This roughness indicates the molecularly imprinted cavities designed for selective binding to the template molecules. The surface roughness morphology of the CuO/g-C_3_N_4_/MIP electrode was verified by the FESEM image, which highlighted cavities formed during the imprinting process, as indicated by the red circle line. In contrast, CuO/g-C_3_N_4_/NIP showed an almost flat polymer matrix, as presented in Figure 7a,b. Moreover, EDS mapping analysis demonstrated a dispersed distribution of C, N, Cu, and O elements on the surface of the CuO/g-C_3_N_4_/MIP nanocomposites. Thus, this observation confirmed the successful dispersibility of CuO/g-C_3_N_4_ within the material in the MIP (molecularly imprinted polymer) matrix, as presented in Figure 7c.

### 3.3. The Electrochemical Behaviors of Modified Electrodes

The electrochemical behaviors of modified electrodes were compared for differences among the bare electrode (CPE), non-imprinted polymer electrode (NIP), molecularly imprinted polymer electrode (MIP), CuO/g-C_3_N_4_-modified electrode, CuO/g-C_3_N_4_/NIP-modified electrode, and CuO/g-C_3_N_4_/MIP-modified electrode. The comparative analysis was conducted using the cyclic voltammetry mode in a solution containing 0.05 M potassium ferrocyanide [K_4_Fe(CN)_6_] with 0.1 M PBS (pH 7.4 at 25 °C), an applied potential in a range from −1 V to 1 V, and a redox reaction presented as shown in Figure 8a. This comparative analysis allows for assessing distinct electrochemical responses associated with surface-modified electrodes. The changed current signals were observed when comparing the various modified electrodes. The highest current signal was demonstrated with the CuO/g-C_3_N_4_ electrode, followed by CPE, CuO/g-C_3_N_4_ /MIP, CuO/g-C_3_N_4_ /NIP, MIP, and NIP electrodes, respectively. Because of the template elution process occurring in the MIP electrode, which generated cavities with a higher surface area and facilitated charge transfer, the results indicated that the current of the MIP electrode was higher than that of the NIP electrode.

On the other hand, the NIP electrode demonstrated a lower current due to its smooth and thick composition. At the same time, the combination of CuO/g-C_3_N_4_ and the MIP for a modified electrode was found to increase current. The observed improvement is due to CuO/g-C_3_N_4_’s advantageous electrical conductivity and capacity to generate electron/hole pairs, which enhance carrier mobility and facilitate charge transfer. The synergistic properties of CuO/g-C_3_N_4_ contribute to the overall improvement in the electrochemical performance of the modified electrode.

Furthermore, the CuO/g-C_3_N_4_/MIP electrode was examined with melamine using the DPV method within the voltage range of −0.1 to −0.3 V under constant light intensity 3 mW/cm^2^. At varied concentrations of melamine, the differential pulse voltammetry response displayed obvious signals, with the switching potential scan towards more negative values indicative of a reduction reaction. Specifically, at an applied potential of −0.2 voltage, a cathodic peak signal was observed. As depicted in Figure 8b, the rise in melamine concentration coincided with a reduction in the current of the cathodic signal. Throughout the amperometry analysis, a consistent potential of −0.2 voltage was applied and maintained. In Figure 8c, to compare various modified electrodes, including NIP, MIP, CuO/g-C_3_N_4_/NIP, and CuO/g-C_3_N_4_/MIP electrodes, an amperometry analysis was performed in PBS as the baseline for 300 s and 100 nM melamine solution for 300 s. The CuO/g-C_3_N_4_/MIP electrode exhibited the highest cathodic current signal. In contrast, NIP and MIP electrodes displayed lower cathodic current signals due to the insulating properties of the polymer matrix hindering charge transfer at the electrode. Testing the CuO/g-C_3_N_4_/MIP electrode for melamine revealed a decrease in the cathodic current signal. This decrease was attributed to the recognition of melamine by the cavities in the molecularly imprinted polymer, impeding charge transfer. In contrast, when the CuO/g-C_3_N_4_/NIP electrode, which was without the template in the modification process, was tested for melamine, there was almost no change in the current signal. Moreover, the CuO/g-C_3_N_4_/MIP electrode demonstrated a more significant delta signal between the baseline and melamine detection, indicating enhanced charge transfer promoted by CuO/g-C_3_N_4_ and the recognition of melamine.

### 3.4. Effect of Light Irradiation

The CuO/g-C_3_N_4_/MIP electrode utilized a photocatalyst material in conjunction with the MIP for melamine detection through photoelectrochemical techniques. The CuO/g-C_3_N_4_ nanocomposite exhibited a narrow energy band gap, rendering it active under visible light. In Figure 9a, the electrochemical behavior of the CuO/g-C_3_N_4_/MIP electrode was tested with various melamine concentrations using the amperometry mode, fixed with an applied potential of −0.2 V, and tested for the presence or absence of light irradiation with a consistent light intensity of 3 mW/cm^2^. The absence of light irradiation resulted in a low cathodic current signal, while the presence of light irradiation showed a high cathodic current signal across all melamine concentrations. Under visible light, the CuO/g-C_3_N_4_ photocatalyst material facilitates accelerated charge transfer and the separation of photogenerated electrons and holes. This process effectively suppresses their recombination, ultimately enhancing the photoelectrochemical performance. Thus, this observation underscores the significant influence of light irradiation on the photoelectrochemical response of the CuO/g-C_3_N_4_/MIP electrode in melamine detection.

### 3.5. Performance of CuO/g-C_3_N_4_ /MIP Electrode

Under light irradiation, the CuO/g-C_3_N_4_/MIP-modified electrode’s electrochemical performance was assessed utilizing electrochemical methods. In Figure 9b, the CuO/g-C_3_N_4_/MIP-modified electrode was tested with various concentrations of melamine; it was observed that higher concentrations of melamine resulted in a decrease in the cathodic current signal. This decrease is attributed to the recognition of melamine by the cavities of the MIP, hindering charge transfer and leading to a reduction in the cathodic current signal. Moreover, the signal analysis between the baseline and melamine detection revealed that the delta current increased with higher concentrations. Thus, the concentration-dependent response further emphasizes the specificity of the MIP in detecting melamine and its impact on the photoelectrochemical behavior of the CuO/g-C_3_N_4_/MIP electrode. The signal for melamine detection using the CuO/g-C_3_N_4_/MIP electrode was calculated by subtracting the baseline signal, and the resulting delta current was plotted against melamine concentration to generate a calibration curve, as demonstrated in Figure 9c. The performance of the CuO/g-C_3_N_4_/MIP electrode for melamine detection using electrochemical techniques under light irradiation was demonstrated by showing the relationship between current and melamine concentration. This examination was conducted within the concentration range of 2.5–1000.0 nM.

As the melamine concentration increased, there was a corresponding elevation in ΔI. However, as the concentration reached 100.0 nM, the signal value exhibited only marginal changes. The calibration plot exhibited a linear relationship between the melamine concentration (x) and the delta current (y), with a range of 2.5–50.0 nM. The linear equation y = 0.0034x + 0.0347 accurately describes this relationship with a correlation coefficient of 0.990. The sensitivity for melamine detection was determined as 4.17 nA/nM, the limit of detection (LOD) was found to be 0.42 nM, and the limit of quantitation (LOQ) was 1.42 nM. Generally, the LOD may be computed as LOD = 3.3 × (S.D.)/M and LOQ = 10 × (S.D.)/M, where S.D. stands for the response’s standard deviation, and M is the calibration curve’s slope.

CuO/g-C_3_N_4_ is a photoactive material, being excited under visible irradiation or suitable energy, leading to the generation of electron/hole pairs. Specifically, the generated electrons at the conduction band of g-C_3_N_4_ migrate to the conduction band of CuO. Subsequently, the photogenerated electrons are transferred to an electrolyte electron acceptor. Concurrently, the electrode provides electrons to neutralize the holes within the excited photoactive materials, resulting in the production of a cathodic photocurrent [30,64,65].

Thus, the CuO/g-C_3_N_4_ mechanism as a photoactive material enhances the facilitation of rapid charge transfer and the separation of photogenerated electrons and holes. This effectively suppresses their recombination, enhancing the photoelectrochemical (PEC) performance, including high sensitivity, a low limit of detection, and a broad dynamic range. Regarding the detection of melamine, the process led to a decrease in the cathodic current signal due to the hindrance of charge transfer between the electrode and the solution caused by the recognition between melamine and the cavities of the MIP, as shown in Figure 10.

### 3.6. Reusability and Reproducibility

The reusability and repeatability of the CuO/g-C_3_N_4_/MIP electrode were evaluated. The CuO/g-C_3_N_4_/MIP electrode’s reusability was assessed through twelve-cycle tests using a melamine concentration of 25 nM, 50 nM, and 100.0 nM, as depicted in Figure 9c. It exhibited the ability to be reused for a total of eight cycles. During the subsequent cycles, the modified electrode displayed a significant rise in the standard deviation signal when detecting melamine. The decrease in current observed in the ninth cycle was due to the repeated elution of melamine for the next cycle’s detection, which resulted in polymer defects like cracking and swelling [66].

Table 1 displays the reproducibility of the CuO/g-C_3_N_4_/MIP electrode, showcasing six electrodes that closely align with the sensitivity, limit of detection, and limit of quantitation values. The CuO/g-C_3_N_4_/MIP electrode fabrication involved meticulous control over both time and temperature. Additionally, it included the precise regulation of the concentration and volume of components, coupled with maintaining a fixed dimension size using Kapton tape for the modified electrode. Therefore, this procedure can be utilized to create the CuO/g-C_3_N_4_/MIP electrode for the purpose of detecting melamine utilizing electrochemical methods under light irradiation.

### 3.7. The Selectivity of the CuO/g-C_3_N_4_/MIP Electrode for Melamine Detection

The modified electrodes with CuO/g-C_3_N_4_ combined with the MIP were constructed and subjected to selectivity testing, which involved the presence of interfering substances such as glycine, uric acid, thiourea, ammonia, urea, and melamine. The CuO/g-C_3_N_4_/NIP electrode demonstrated a weak signal across all tests. The CuO/g-C_3_N_4_/MIP electrode demonstrated a strong and reliable signal response, particularly in the identification of melamine, as illustrated in Figure 11a. In contrast, other interferents displayed a comparatively weak signal. This observation underscores the excellent selectivity of the CuO/g-C_3_N_4_/MIP electrode for melamine detection. The selectivity of this process was attributed to the molecularly imprinted polymer (MIP) approach, in which the removal of the template molecule creates particular cavities with molecular memory. The design of these cavities is customized to specifically target melamine molecules, considering their distinctive features such as size, shape, and functional groups. Melamine molecules, which are nitrogen-rich heterocyclic triazine compounds, form interactions with the acidic carboxyl group of the functional monomer MAA. This interaction is facilitated through mechanisms like van der Waals forces, electrostatic interactions, and hydrogen bonding, owing to the specific affinity between the carboxyl group of the MAA monomer and the amino groups of melamine. The particular affinity of the resulting CuO/g-C_3_N_4_/MIP electrode greatly enhances its capacity to selectively detect melamine [51,67,68].

### 3.8. Real Sample Analysis

The CuO/g-C_3_N_4_/MIP electrode was employed for melamine detection using photoelectrochemical techniques and was tested with actual samples such as cow milk, strawberry milk, and almond milk. Additionally, melamine-contaminated variants of cow milk, strawberry milk, almond milk, and pure melamine are presented in Figure 11b. The signal response was examined for the melamine-contaminated versions of cow milk, strawberry milk, almond milk, and pure melamine, revealing a significant signal. In contrast, non-melamine-contaminated samples exhibited a low signal. Therefore, it can be deduced that the CuO/g-C_3_N_4_/MIP electrode exhibits both high sensitivity and selectivity for detecting melamine.

The comparison of the modified CuO/g-C_3_N_4_/MIP electrode for melamine detection with alternative methods is shown in Table 2. Although HPLC-MS/MS exhibits high selectivity and a notable detection limit, its long operational time and complexity limit its practical use for users [6]. Quartz crystal microbalances modified with molecularly imprinted polymers (MIPs) exhibited high selectivity [67]. Nevertheless, this method is challenging to control reactions due to its sensitivity to mass changes and susceptibility to environmental perturbations. The fluorescent method, employing label-free Thymine/SYBR Green I, demonstrated notable sensitivity and selectivity. However, it is high-cost, requires complexity in preparation, and has potential sample color interference [10]. On the other hand, the electrochemical approach was selected for melamine detection due to its numerous advantages, including a low detection limit, high sensitivity, rapid response time, and compatibility with portable melamine detection devices.

Moreover, incorporating nanomaterial properties further enhanced the sensitivity of melamine detection by facilitating the charge transfer process. Advanced materials, such as poly (para-aminobenzoic acid)/GCE [69], GCE/P-Arg/ErGO–CuNFs [70], and CuO/MIP [71], were used to improve sensitivity. However, these substances proved costly and required a complex synthesis process. In contrast, CuO/g-C_3_N_4_/MIP significantly enhanced melamine detection performance by integrating photoactive materials and electrochemistry. This approach not only achieved high sensitivity but also presented a low limit of detection, high selectivity, cost-effectiveness, ease of preparation, and the potential for development into a portable device.

## 4. Conclusions

In summary, the hetero-structure is endowed with a narrower band gap (1.40 eV) as a result of the effective integration of g-C_3_N_4_ with CuO. This improvement enhances the efficiency of electron/hole pair separation, thereby enhancing the photoelectrochemical (PEC) signal under visible light. In addition, the modification of the CuO/g-C_3_N_4_ nanocomposite in conjunction with the molecularly imprinted polymer was evaluated for melamine detection using electrochemical techniques under visible light. The modified electrode demonstrated excellent sensitivity, a low limit of detection of 0.42 nM, and a low limit of quantitation of 1.42 nM. These values are lower than the melamine levels that the FDA and WHO have established for food products (TDI levels of 0.2 mg/kg).

Additionally, the electrode demonstrated a large linear range, high repeatability, and eight cycles of melamine detection reusability. Furthermore, the modified electrode demonstrated elevated selectivity in melamine detection, attributable to the affinity cavities created by the molecularly imprinted polymer (MIP). The CuO/g-C_3_N_4_/MIP-modified electrode was designed specifically for the photoelectrochemical detection of melamine, with the possibility for future development into a portable device. This innovation is well suited for applications related to the detection of food contamination.

## Figures and Tables

**Figure 1 polymers-16-01800-f001:**
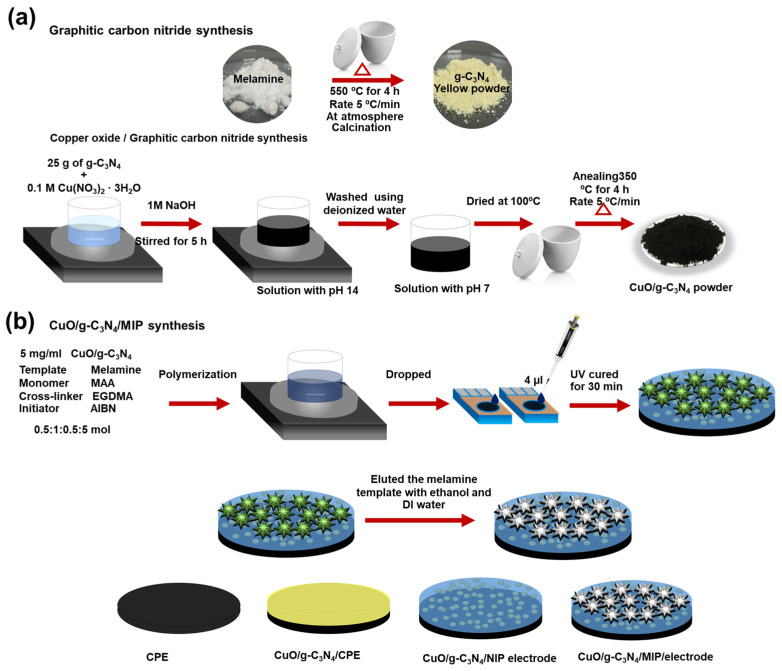
Schematic illustration of (**a**) CuO/g-C_3_N_4_ synthesis and (**b**) combination CuO/g-C_3_N_4_/MIP synthesis for modified electrode.

**Figure 2 polymers-16-01800-f002:**
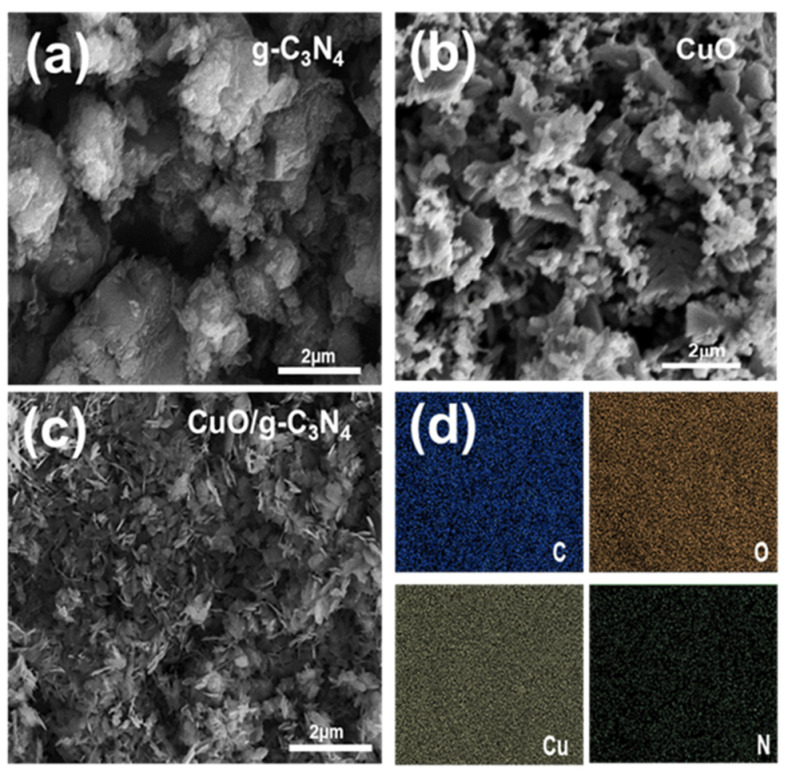
FESEM images of (**a**) pure g-C_3_N_4_, (**b**) CuO, and (**c**) CuO/g-C_3_N_4_ nanocomposites and (**d**) EDS element mapping of C, Cu, O, and N.

**Figure 3 polymers-16-01800-f003:**
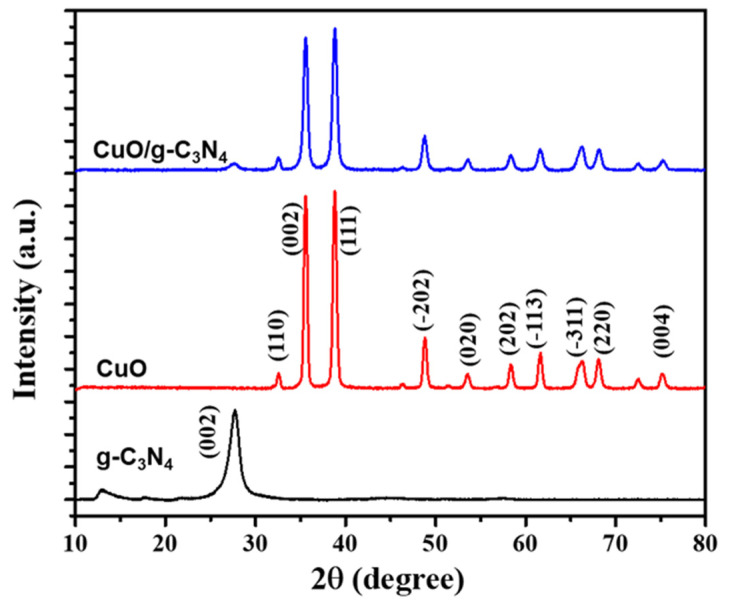
XRD pattern of pure g-C_3_N_4_, CuO, and CuO/g-C_3_N_4_ nanocomposites in range of 10° to 80°.

**Figure 4 polymers-16-01800-f004:**
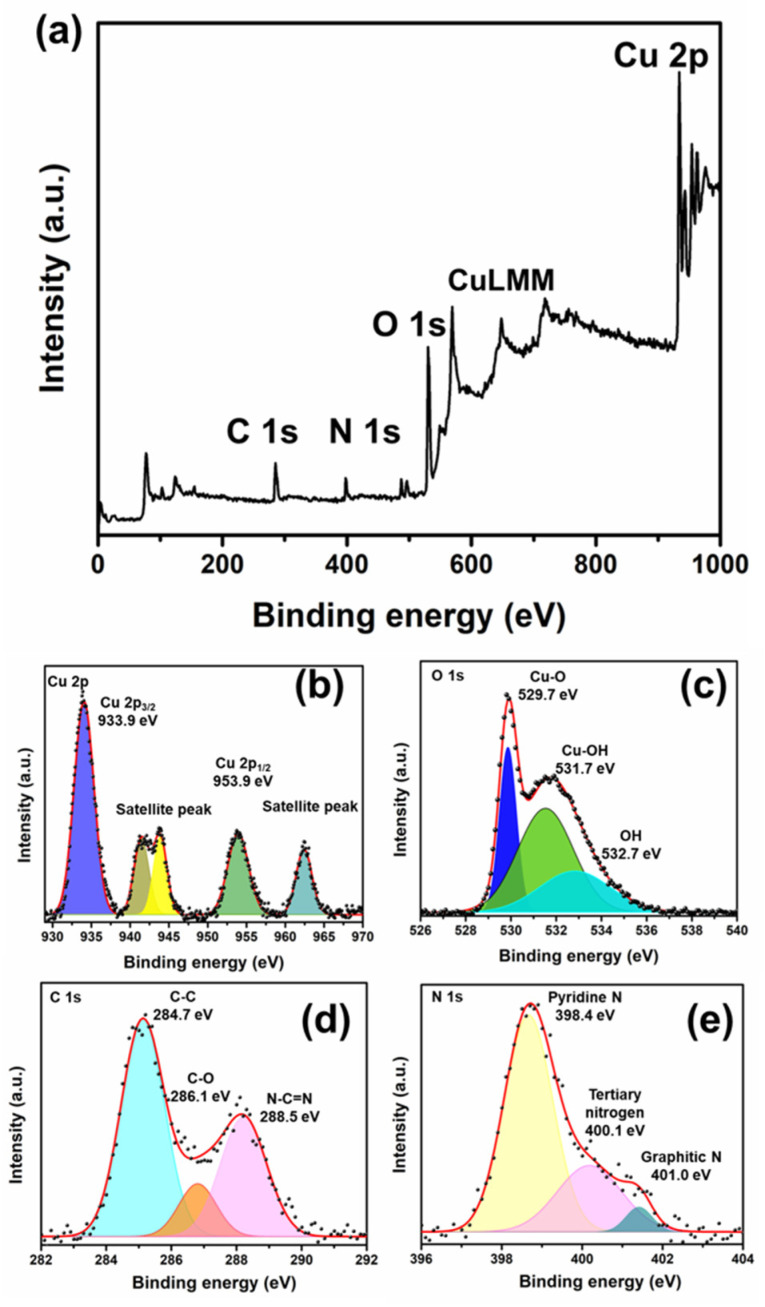
XPS spectra of the synthesized CuO/g-C_3_N_4_ nanocomposites (**a**) sample survey and the high-resolution XPS spectra for (**b**) Cu 2p, (**c**) O 1s, (**d**) N 1s, and (**e**) C 1s.

**Figure 5 polymers-16-01800-f005:**
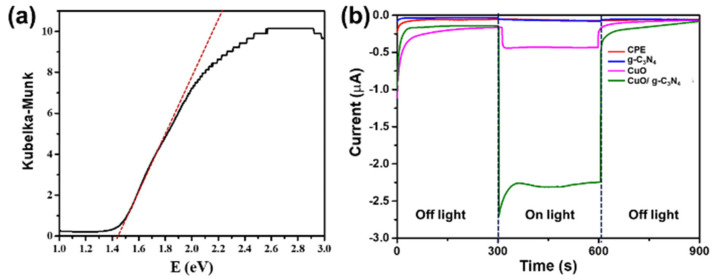
(**a**) Kubelka–Munk plot of CuO/g-C_3_N_4_ in red line is intersection and (**b**) amperometry of bare electrode, CuO, g-C_3_N_4_, and CuO/g-C_3_N_4_ in PBS solution, applied potential at −0.2 V, and presence or absence of light irradiation.

**Figure 6 polymers-16-01800-f006:**
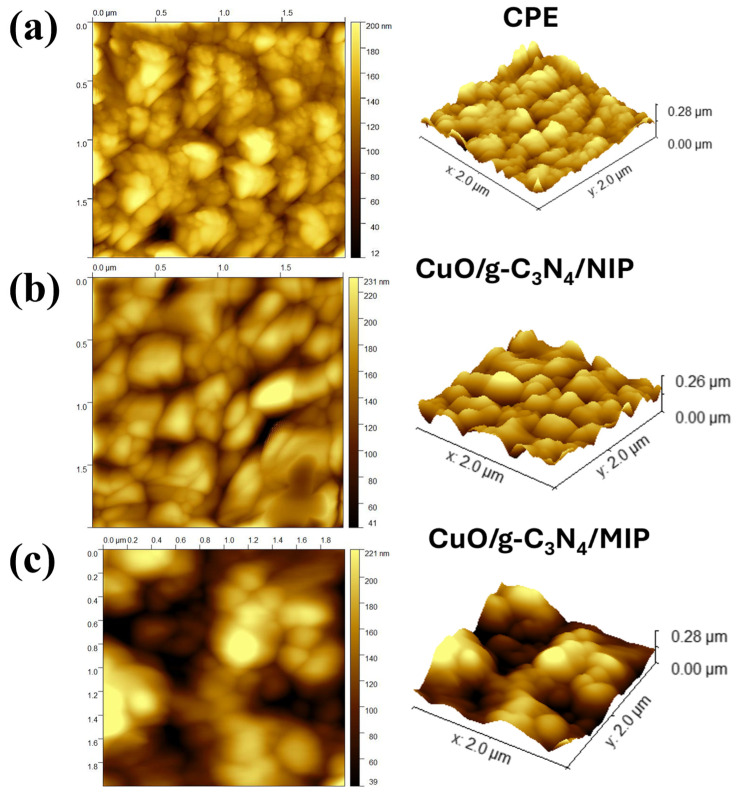
Three-dimensional (3D) AFM images of (**a**) bare electrode, (**b**) CuO/g-C_3_N_4_/NIP, and (**c**) CuO/g-C_3_N_4_/MIP.

**Figure 7 polymers-16-01800-f007:**
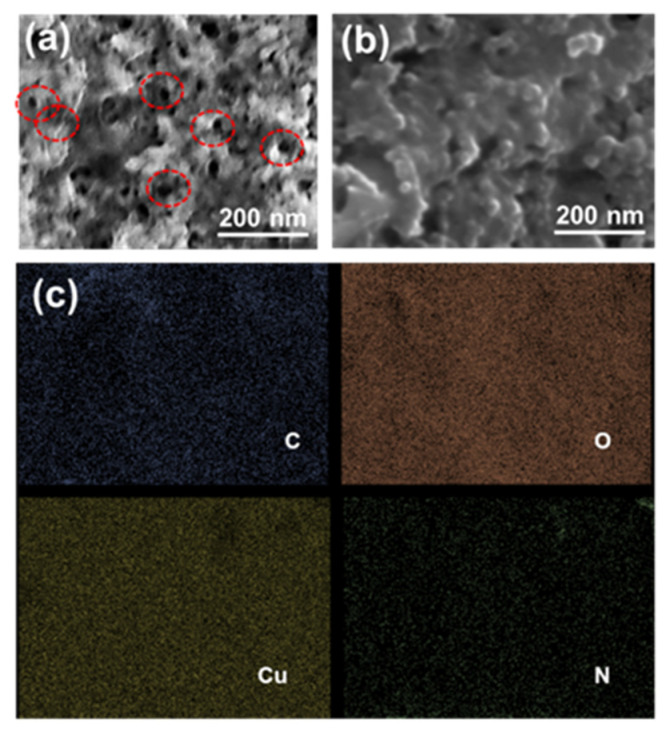
FESEM image of (**a**) CuO/g-C_3_N_4_/MIP, (**b**) CuO/g-C_3_N_4_/NIP, and (**c**) EDS mapping of CuO/g-C_3_N_4_/MIP including C, O, Cu, and N.

**Figure 8 polymers-16-01800-f008:**
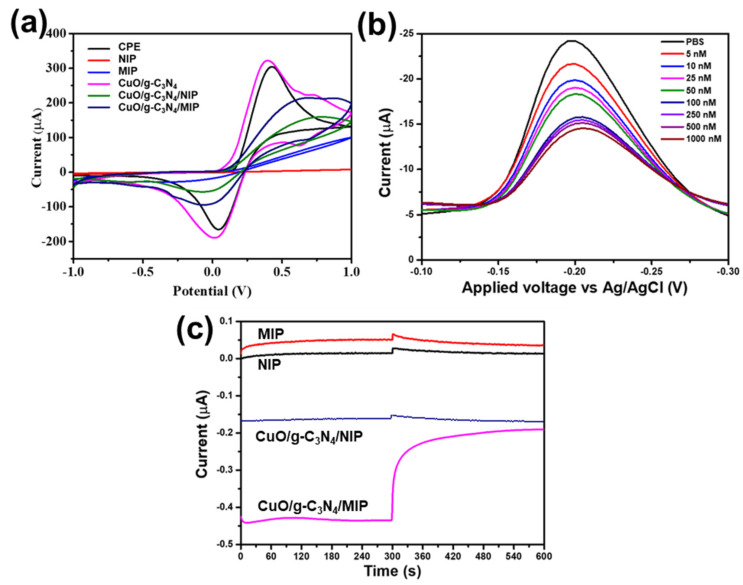
Electrochemical behavior consists of (**a**) cyclic voltammogram of different modified electrodes in 0.05 M [K_4_Fe(CN)_6_] containing 0.1 M PBS (pH 7.4 at 25 °C), (**b**) differential pulse voltammetry of CuO/g-C_3_N_4_ in various melamine concentrations, and (**c**) amperometry of difference electrode in PBS for 300 s and then 100 nM melamine for 300 s (under light irradiation).

**Figure 9 polymers-16-01800-f009:**
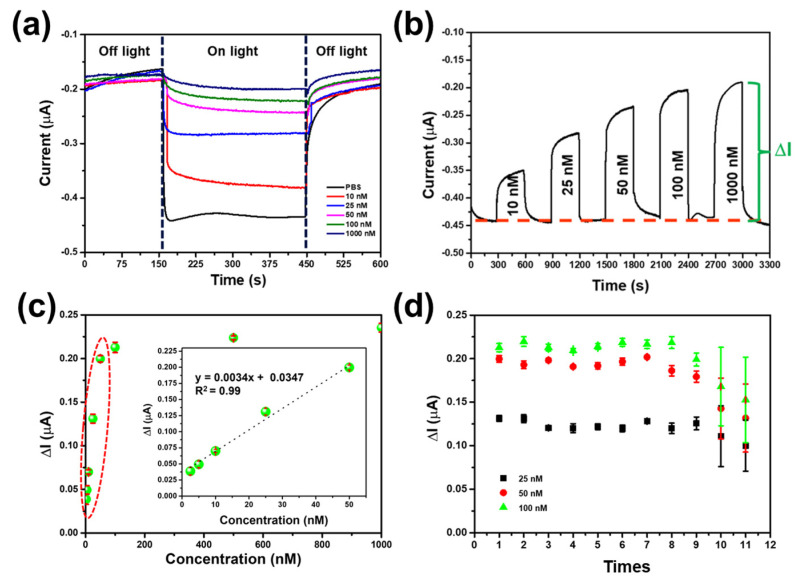
The electrochemical behavior of the CuO/g-C_3_N_4_/MIP electrode consists of (**a**) the effect of the presence or absence of light irradiation, (**b**) the amperometry of various melamine concentrations, (**c**) the response curve and standard calibration plot (inset) of the CuO/g-C_3_N_4_/MIP electrode, and (**d**) reusability for melamine detection.

**Figure 10 polymers-16-01800-f010:**
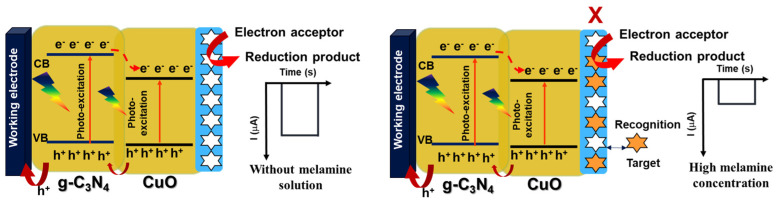
The proposed CuO/g-C3N4/MIP electrode mechanism for melamine detection using photoelectrochemical techniques.

**Figure 11 polymers-16-01800-f011:**
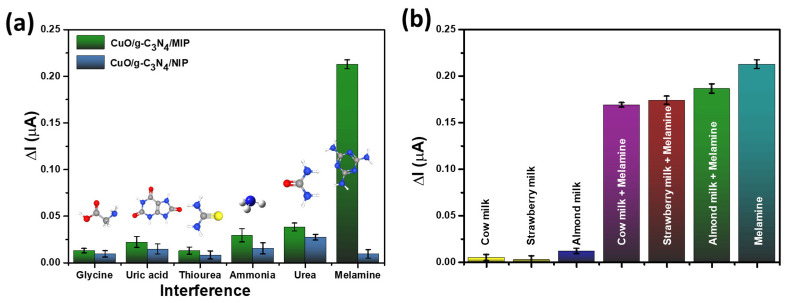
The CuO/g-C_3_N_4_/MIP electrode and the CuO/g-C_3_N_4_/NIP electrode’s response in (**a**) selectivity with interference containing glycine, uric acid, thiourea, ammonia, urea, and melamine and (**b**) testing using real samples, including cow milk, strawberry milk, and almond milk, as well as melamine-contaminated variants of cow milk, strawberry milk, and almond milk.

**Table 1 polymers-16-01800-t001:** CuO/g-C_3_N_4_/MIP electrode’s reproducibility in melamine detection.

Electrode	Linear Equation	R^2^	Sensitivity (nA/nM)	LOD (nM)	LOQ (nM)
1	y = 0.0036x + 0.034	0.990	4.17	0.42	1.42
2	y = 0.0032x + 0.032	0.991	4.29	0.30	1.02
3	y = 0.0033x + 0.039	0.995	4.69	0.29	0.97
4	y = 0.0033x + 0.031	0.992	3.81	0.56	1.89
5	y = 0.0033x + 0.027	0.992	3.45	0.42	1.40
6	y = 0.0049x + 0.023	0.992	4.37	0.24	0.80

**Table 2 polymers-16-01800-t002:** Comparing the modified CuO/g-C_3_N_4_/MIP electrode for melamine detection with alternative methods.

Methods	Materials	Linear Range (nM)	LOD (nM)	Ref.
HPLC–MS/MS	-	200–5000	50	[6]
Quartz crystal microbalances	MIP	1 × 10^3^–3 × 10^5^	8	[67]
Fluorescence	Thymine/SYBR Green I	1 × 10^4^–2 × 10^6^	158	[10]
Electrochemistry	poly(para-aminobenzoic acid)/GCE	4.0 × 10^3^–450 × 10^3^	36	[69]
Electrochemistry	GCE/P-Arg/ErGO–CuNFs Electrode	10–90	5.0	[70]
PEC	CuO/MIP	5–50	2.45	[71]
PEC	CuO/g-C_3_N_4_/MIP	2.5–50	0.42	This work

## Data Availability

Data are contained within the article.

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
