# Peer review of "Enhancement in Sensitivity and Selectivity of Electrochemical Technique with CuO/g-C3N4 Nanocomposite Combined with Molecularly Imprinted Polymer for Melamine Detection"

_polymers, 2024, doi:10.3390/polym16131800_

Round 1
Reviewer 1 Report
Comments and Suggestions for Authors
Sentence “A World Health Organization (WHO) expert meeting report examined cyanuric acid and melamine's health effects and toxicological characteristics.” Can you please provide a refernce for this report?
g-C3N4 – the 3 and 4 should be written as subscripts everywhere.
The quality of pictures is very pure.
There was reported a mistake in the previous works regarding Kubelka-Munk method. By this reason I recommend to add this work to confirm that you consider this fakt and use the right way: This allowed the calculation of the optical band gap through the Kubelka–Munk equation [61; DOI: 10.1088/1402-4896/ad1cb8].
Figure 6 and 8: The scales are invisible.
It is not possible to read the information at figure 10 and 11.
Could you please provide the valence band maximum measurement from XPS?
Comments on the Quality of English LanguageThe language should be improved.
Author Response
Dear reviewer,
We appreciate the reviewer for your valuable time in reviewing our article and providing valuable comments. Your valuable and insightful comments led to possible improvements in the current version. The authors have carefully considered the comments and tried our best to address every one of them. The grammar and style of the English language have been corrected. We hope the manuscript, after careful revisions, meets your high standards. The authors welcome further constructive comments if any. Below, we provide the point-by-point responses. All modifications in the manuscript have been highlighted.

Reviewer 2 Report
Comments and Suggestions for Authors
The work is devoted to development of the system for electrochemical detection of melamine in solution. Melamine is an important contaminant, which poses health risks to humans. Thus, producing devices for its easy and convenient detection is timely. The authors propose to use for this purpose a complicated system: CuO/g-C3N4 nanocomposite combined with molecularly imprinted polymer. They successfully demonstrated electrochemical response of this system and proved that this response depends on the melamine concentration. They claimed that decrease of delta-current at higher melamine concentration is caused by binding of melamine molecules into the pores in polymer produced by molecular imprinting. However, this important conclusion is not supported adequately by the experimental data. To do this, the authors should perform melamine sensitivity and selectivity tests for CuO/g-C3N4 nanocomposite combined with polymer without molecular imprinting (CuO/g-C3N4/NIP). The results should be compared with that for CuO/g-C3N4/MIP system. Besides, a number of small drawbacks should be corrected:
Line 27. In the abstract the linear detection range from 2.5 nM to 1000 nM is stated, while in the text this range is 2.5 – 50 nM. The upper limit of this range in the abstract should be changed.
Line 134. Abbreviation RPE should be introduced.
Line 154. Abbreviation MAA should be introduced.
Lines 165 and 166. Abbreviations AIBN and EDGMA should be introduced.
Figure 1. Resolution is too low.
Fig 2d. Scale of EDS maps should be specified.
Fig. 5a. The physical quantity should be specified for y-axis. “Kubelka-Munk” is not a physical quantity.
Text above Fig. 5. What is the precision of bandgap determination? It seems that bandgap for CuO and CuO/g-C3N4 is nearly the same. The synergistic affect in this composite is not evident. Probably, this conclusion should be removed.
Lines 336 – 338. The sentence “Therefore, the nanocomposite was confirmed as CuO/g-C3N4 nanocomposite with the structure of heptazine of g-C3N4 coupling Monoclinic of CuO by XRD pattern and the elemental composition of a material from XPS and FESEM/EDS.” is not understandable. I suggest to remove it and the next sentence.
Line 357. Abbreviation NIP should be introduced. Preparation of NIP should be described in the text.
Fig. 6 and the text below it. The roughness of MIP (more than 100 nm) is substantially larger than the size of melamine molecule (several nm). The authors should explain why molecular imprinting produced such a huge roughness.
Line 406. Physical unit (V) should be added at this part “voltage range of -0.1 to -0.3”
Fig. 8 is too crowded, and its resolution is too low. The “light on” and “light off” intervals should be specified at Fig. 8c
Lines 471 – 474. Comparison of LOD for melamine and FDA and WHO thresholds is incorrect. These thresholds are related to the intake of melamine per day, not to its allowed concentration in solution. This comparison should be removed.
Comments on the Quality of English LanguageTitle should be corrected into “Enhancement of Sensitivity and Selectivity of Electrochemical Technique with CuO/g-C3N4 Nanocomposite combined with Molecularly Imprinted Polymer for Melamine detection.”
Lines 17 – 20. The sentence is too long and cannot be understood. It should be split into 2 sentences.
Line 22. “analyzed through” should be changed into “performed using”
Line 42. “European member states” should be changed into “European Union member states”
Line 119. The phrase “methacrylic acid was employed as the monomer to polymerize the melamine-imprinted polymer” should be changed into “methacrylic acid was employed as the monomer to polymerize forming the melamine-imprinted polymer”
Lines 214 – 216. The sentence “For the selectivity study, interference comparisons were conducted with consistent concentrations of 100.0 nM for each thiourea, ammonia, glycine, urea, uric acid, and melamine solution.” should be changed into “For the selectivity study, interference comparisons were conducted for each of thiourea, ammonia, glycine, urea, uric acid, and melamine solution with consistent concentrations of 100.0 nM.”
Author Response
Dear reviewer,
We appreciate the reviewer for your valuable time in reviewing our article and providing valuable comments. Your valuable and insightful comments led to possible improvements in the current version. The authors have carefully considered the comments and tried our best to address every one of them. The grammar and style of the English language have been corrected. We hope the manuscript, after careful revisions, meets your high standards. The authors welcome further constructive comments if any. Below, we provide the point-by-point responses.

Reviewer 3 Report
Comments and Suggestions for Authors
The article can be evaluated positively overall, but it has some shortcomings and contains some suggestions for Authors errors that should be addressed before it is forwarded for publication.
1. The pictures in the paper are blurry, and it is highly recommended to improve the clarity of the pictures in the paper.
2. In lines 19, 90, 496, 509, and 576, C3N4 is not changed to subscript.·
3. None of the chemical formulas in the title of the reference paper have been subscripted, and it may be better if they were changed to subscript.
Regarding all the above comments with all minor recommendations, the paper can be accept in the revised form for publication.
Comments on the Quality of English LanguageThe article can be evaluated positively overall, but it has some shortcomings and contains some suggestions for Authors errors that should be addressed before it is forwarded for publication.
1. The pictures in the paper are blurry, and it is highly recommended to improve the clarity of the pictures in the paper.
2. In lines 19, 90, 496, 509, and 576, C3N4 is not changed to subscript.
3. None of the chemical formulas in the title of the reference paper have been subscripted, and it may be better if they were changed to subscript.
Regarding all the above comments with all minor recommendations, the paper can be accept in the revised form for publication.
Author Response

(The authors gave the same response as above.)

Round 2
Reviewer 1 Report
Comments and Suggestions for Authors
The manuscript was corrected and can be published.
Comments on the Quality of English LanguageTypos should be corrected during the proof reading.
Reviewer 2 Report
Comments and Suggestions for Authors
The manuscript has been imprived substantially upon revision. I do not have further suggestions for revision.